# Chess-like Pieces Realized by Selective Laser Sintering of PA12 Powder: 3D Printing and Micro-Tomographic Assessment

**DOI:** 10.3390/polym16243526

**Published:** 2024-12-18

**Authors:** Giovanna Colucci, Luca Fontana, Jacopo Barberi, Chiara Vitale Brovarone, Massimo Messori

**Affiliations:** 1Department of Applied Science and Technology (DISAT), Politecnico di Torino, Corso Duca Degli Abruzzi 24, 10129 Torino, Italy; luca.fontana@polito.it (L.F.); jacopo.barberi@polito.it (J.B.); chiara.vitalebrovarone@polito.it (C.V.B.); massimo.messori@polito.it (M.M.); 2National Interuniversity Consortium of Materials Science and Technology (INSTM), Via G. Giusti 9, 50121 Firenze, Italy; 3Centro Interdipartimentale Polito BioMEDLab, Politecnico di Torino, Corso Duca degli Abruzzi 24, 10129 Turin, Italy

**Keywords:** 3D printing, additive manufacturing, selective laser sintering, micro-tomography, polyamide 12

## Abstract

The paper highlights the realization of 3D-printed parts with complex geometries, such as chess-like pieces, using polyamide 12 (PA12) as polymeric powder via selective laser sintering (SLS). The research activity focuses on the study of the powder printability, the optimization of the printing parameters, and the tomographic evaluation of the printed objects. Morphological analyses were carried out to study the PA12 powder microstructure considering that SLS required specific particle size distribution and shape, able to guarantee a good flowability necessary to take part in a sintering process. DSC and TG analyses were performed to determine the sintering window and the crystallinity degree, and to evaluate the thermal stability of the PA12 powder due to the importance of the powder processability for the SLS process. The novelty lies in the realization of chess-like pieces very challenging to print via SLS due to their different and highly detailed structures, and the in-depth analysis of the dimensional accuracy evaluated by micro-tomography. The 3D-printed samples obtained show high printing quality and dimensional stability. The μ-CT analysis also confirms the key role of the object shape and section changes on the final porosity of the chess-like pieces.

## 1. Introduction

The key advantage of additive manufacturing (AM) is the possibility to realize complex components by using rapid prototyping, reducing the process time, process steps, raw materials, wastes, and costs [1,2]. AM uses 3D modelling to develop the required design and different kinds of 3D printers adapted to optimize the manufacturing process and to produce the desired geometry of the final products, which are generally realized within the least possible time by adding successive layers of material on top of each other until the process is completed [1,2].

Researchers from industry are widely using AM to develop mass customization of products, reducing the time in critical replacement of some parts that need to be repaired and shortening the supply chain, which is not possible by any other conventional manufacturing processes. The high flexibility of design of 3D printing further helps the end consumers use the AM technology with ease. In the last few years, large companies in automotive and aerospace have been switching to AM for manufacturing parts, reducing the dependence on external service providers, particularly for consumers from remote locations [1,2,3,4]. Moreover, AM also leads to reduced cost of transportation by avoiding stockpiling, which indirectly influences the final price of the component and, as a consequence, the material wastage, achieving benefits from both economic and environmental points of view [5,6].

In contrast, small and medium enterprises still hesitate to work with AM due to the large investment in the setup of the 3D printing machines and lack of operating knowledge. However, due to the aforementioned advantages of AM, the overall cost of the products developed can be considerably lower compared to conventional manufacturing processes.

Different AM technologies are available to process thermoplastic polymers, such as fused filament fabrication (FFF) [7,8,9,10,11,12,13,14] and selective laser sintering (SLS) [15,16,17,18,19] for thermoplastics, that result in products with complex geometries and good dimensional resolution with enhanced thermal and mechanical properties by optimizing the printing parameters. SLS can be considered the most consolidated laser powder bed fusion (LPBF) AM technology for polymer processing because it allows the manufacture of parts with high structural complexity, dimensional accuracy, and surface quality, without the need for supports or molds since the unsintered powder surrounding the consolidated cross-section acts as a support, and without further post-processing steps [15,16,17,18,19].

3D-printed objects can be obtained by means of SLS from a layer-by-layer sintering of fine powders due to the action of a laser that selectively scans the powder bed according to a digitally defined design involving the use of thermoplastic polymeric powders with characteristic shape and dimension. Specifically, the powder should be as spherical as possible to guarantee free-flowing properties during spreading and have a particle size below 100 µm [17,18,19,20].

Many studies are reported in literature on the use of polyamide powders in SLS processing [20,21,22]. Polyamides are semicrystalline thermoplastic polymers that include polyamide-6 (PA6) [23,24], polyamide-11 (PA11), [25] and polyamide-12 (PA12) as such or in their reinforced form [26,27,28,29,30,31,32]. Among them, PA12 is the most used polymer in SLS, representing around 95% of the total polymer market due to its high versatility and good properties such as toughness and heat and chemical resistance, which allows PA12 to realize rigid, detailed, and stable parts for long-term use via SLS [26,27,28,29,30,31,32].

The PA parts’ quality and their final properties are strongly affected by the printing parameters, such as the laser power and speed, bed temperature, scan spacing, layer thickness, and orientation. However, it is very important to properly tune all those parameters to prevent instabilities during the sintering process, such as the formation of porosities, generally responsible for the poor properties of final 3D-printed pieces [33].

There is an extensive list of papers in literature related to the study of the effect of the presence of voids or pores in the structures of PA12 products fabricated by fused filament fabrication process [12,13,14].

Many studies have examined the relation between the process parameters, such as laser power, scan speed, layer thickness, and preheating temperature, and the part properties in terms of tensile strength, modulus, elongation at break, and density. The aim was to extend process knowledge on SLS of polymers, increasing the process control [28,29,30,31]. Other authors have explored how the process parameters strongly influence the structures containing voids, correlating this aspect with their final properties. As pores are generally considered critical defects, the research seeks to understand how the porosity and the pores’ formation can influence the morphology and the mechanical properties of the printed parts obtained via SLS [32,33,34,35].

Only a few papers report the pore analysis and mechanical properties of parts in PA12 unfilled and filled with carbon fibers obtained by SLS but with very simple geometry, like dog bones and cylinders [36,37]. However, according to the author’s best knowledge, no studies have examined the microscopic internal structure of SLS-processed parts with complex geometries, such as chess-like pieces.

The novelty introduced with this paper lies in the SLS of 3D-printed complex shapes (e.g., chess-like pieces) that are very challenging due to the high variety of shapes and dimensions, and levels of detail.

Moreover, the paper focuses on the study of the porosity effect, object shape, and section changes on the final properties of the 3D parts, quantitatively characterized by X-ray computed tomography. In fact, the microscopic analysis of the internal pores in terms of size, density, and spatial location offers an overview of the whole structure of the consolidated 3D-printed pieces, whereas 3D scanning enabled evaluation of dimensional accuracy of the chess-like pieces realized by SLS with respect to the CAD model selected for the 3D printing process.

## 2. Materials and Methods

### 2.1. Materials and 3D Printing

Polyamide 12 (PA12) used in this work was a commercially available DuraForm^®^ PA12 powder provided by 3D Systems GmbH (Darmstadt, Germany), with density of 1.0 g/cm^3^, as reported by the supplier technical datasheet.

The PA12 powder was processed using a Sharebot SnowWhite2 SLS machine (Sharebot S.r.l., Nibionno, Italy), which uses a CO_2_ laser having a 14 W maximum power (λ = 10.6 μm) to selectively fuse in air the polymeric powder deposited in thin successive layers by means of a recoating blade over the build plate.

The process parameters for the realization of the SLS parts were optimized to minimize the porosity content, avoid part distortion, and reach good printing resolution.

The optimization of the printing parameters was done by means of a comprehensive experimental testing procedure. Initially, the printability of the PA12 powder was assessed starting from printing parameters reported from previous literature on similar works [17,18,19]. Then, a systematic optimization of the relevant settings for the realization of the chess-like pieces was carried out. The printing parameters were changed every single print until the best resolution details were obtained.

At the end, the powder bed temperature was set at around 166–170 °C, the laser power at 4.6 W, the scanning speed at 3300 mm/s. The layer thickness was fixed at 0.1 mm, while the infill density was fixed at 100% for all the parts to ensure very dense structures. Twenty warming layers were also used for each print, i.e., spread for twenty consecutive layers, without any laser hatching.

### 2.2. Characterization Techniques

The morphology of the PA12 powder was investigated using a Phenom™ XL Scanning Electron Microscope (Waltham, MA, USA) at a voltage of 15 kV after metallization with platinum.

The particle size distribution (PSD) of PA12 powder was performed by examining several thousands of particles using an automated analyser Morphology 4 (Malvern Panalytical, Malvern, UK) to obtain a statistical evaluation of the particle size and shape of the polymer powder. It employs a sophisticated image-processing software to capture high-resolution images of each particle and analyses them based on shape features such as aspect ratio, circularity, and elongation.

The true density (ρ) of the PA12 powder was evaluated using an Ultrapyc 5000 gas pycnometer (Anton Paar GmbH, Graz, Austria) using helium as probe gas at room temperature, according to the standard ASTM B923-20, to characterize the flowability of the powder. Three consecutive measurements, within a tolerance of 0.005%, were chosen to improve the accuracy of the experimental results.

Thermogravimetric analysis (TG) was carried out on the PA12 powder and on the 3D-printed samples using a Mettler-Toledo TGA 851e Instrument (Columbus, OH, USA), from 25 to 900 °C with a heating rate of 10 °C/min and 50 mL/min in air.

Differential scanning calorimetry (DSC) was performed on the PA12 powder and on the 3D-printed samples using a Netzsch 214 Polyma Equipment (Selb, Germany) from −50 to 250 °C with a heating/cooling rate of 10 °C/min with a nitrogen flow of 40 mL/min. Two heating scans and one cooling scan were done.

The degree of crystallization (*X_c_*) was estimated by using Equation (1):(1)$$X_{c}=\frac{{∆H}_{m}}{{∆H}_{m}^{0}}\times 100$$
where ∆*H_m_* and ∆*H_m_*^0^ represent the experimental enthalpy of melting and the enthalpy of fusion of fully crystalline PA12 (209 J/g), respectively [36,37].

The bulk density of the 3D-printed samples was also evaluated through the non-destructive Archimedes method according to the standard ASTM B962–17. Since the tested material has density close to 1 g/cm^3^, isopropyl alcohol (ρ = 0.785 g/cm^3^) was used as liquid instead of distilled water. Five different specimens were tested to obtain statistically significant results.

The porosity was then calculated following Equation (2):(2)$$P=\frac{\rho -\rho_{sls}}{\rho }\times 100$$
where ρ*_sls_* is the density of each 3D-printed sample and ρ is the true density of the starting PA12 powder previously measured using the gas pycnometer [17,18].

The ρ*_sls_* density values of the 3D-printed chess-like pieces were determined by using Equation (3) considering the total porosity present in the material (Total porosity density):(3)$$\rho_{sls}=\frac{W_{air}*\rho_{liq}}{W_{fin}-W_{liq}}$$
and Equation (4) considering the closed porosity (Closed porosity density):(4)$$\rho_{sls}=\frac{W_{air}*\rho_{liq}}{W_{air\;}-W_{liq}}$$
where *W_air_* is the mass of the sample in air, *W_liq_* is the mass of the sample immersed in the liquid, *W_fin_* is the mass of the sample in air after being extracted from the liquid, and ρ*_liq_* is the density of the isopropyl alcohol in which the samples were immersed.

X-ray micro-computed tomography (µ-CT) analysis of the 3D-printed samples was performed using a benchtop scanner (Skyscan1272, Bruker, Billerica, MA, USA), using an accelerating voltage of 70 kV, a current equal to 160 µA, and a 0.5 mm Al filter in front of the detector to cut the low energy photons. The voxel size was set at 10 µm, and the projections were acquired over 180° with an angular step of 0.2°. The reconstruction of the cross-sections was performed with dedicated software (Nrecon, Bruker, MA, USA), obtaining grayscale images. The reconstructed images were analysed by adapting the workflow proposed by du Plessis et al. [38] to obtain morphometrical parameters using the dedicated software CTan 1.19.11.0 (Bruker, USA), particularly total porosity, open porosity, closed porosity, pore size distribution, and porosity distribution along the printing direction.

Briefly, the cross-section images were binarized, attributing white pixels to the object and black ones to the background, and defects below 8 voxels were removed. The region of interest (ROI) was then wrapped around objects in the 2D cross-section, extending it across voids of 30 pixels to consider surface porosity. In this way, it was possible to evaluate the total porosity of the 3D-printed object and calculate the contribution of the close and the open porosity, respectively. Pore size distribution was evaluated by reversing the binarization to analyse the pores as objects and evaluating their structure thickness.

Printing fidelity was assessed by comparing the 3D reconstruction of the objects with the original CAD models.

The data acquired were then inspected through the GOM Suite Software. The comparison of the scan data and reference data in the GOM Inspect environment was carried out on a computer with Windows 11 installed, 32 GB of RAM, an Intel^®^ Core™ i7-10870H, and clock frequency of 2.20 GHz. The graphic card was an NVIDIA RTX3060. Once the scan data were acquired, the relative STL file was imported in GOM Inspect environment to perform quality and displacement measurements. In particular, the software performed a comparison between the scan data and the reference CAD model used to produce the printed part.

## 3. Results and Discussion

### 3.1. PA12 Powder Characterization

SEM analyses were firstly performed to evaluate the morphology, shape, and size of PA12 powder to determine its processability via SLS. Figure 1 reports SEM micrographs of the powder at different magnifications, 300× (a) and 1000× (b), respectively. From the images, it is possible to see that the polymeric powder is characterized by irregular elongated-shaped particles with a size lower than 80 microns, in line with the required powder size range for optimal SLS processing [17,18,19].

Moreover, to better estimate the particle-size distribution of the PA12 powder, a granulometric analysis was performed by analysing about a thousand particles. Table 1 summarises the diameter, aspect ratio, and circularity of PA12 particles evaluated, in order to determine the distribution of particle sizes within the powder, considering that particles with dimensions lower than 100 microns can be useful for SLS applications.

The results suggest that PA12 powder can be successfully processed in the SLS printer having the ideal particles’ morphology, dimension, and flowability for ensuring the spreading of a defect-free powder layer by layer, leading to the production of well-defined printed parts. Finally, the thermal stability of neat PA12 powder was studied by using TG in air, while the thermal transitions and degree of crystallinity were evaluated via DSC in nitrogen atmosphere.

PA12 thermal degradation consists of one main peak, where the temperature at which the maximum degradation peak occurs, evaluated by the derivative mass loss data, was identified at 438 °C, as visible from Figure 2a.

The cooling and heating curves of PA12 powder generated by DSC analysis from the single cooling and the second heating scans are reported in Figure 2b, evidencing a melting temperature at 175 °C and a crystallization temperature at 145 °C. The glass transition temperature was also found at 60 °C. The T_g_ represents the temperature at which the PA12 transitions from a rigid, glassy state to a more flexible, rubbery state. In the context of our research, T_g_ value was determined to highlight the thermal behavior of PA12, which is relevant for understanding its final performance in different temperature environments, such as within the 3D printer. However, like other semicrystalline polymers, PA12 does not show a sharp, well-defined baseline shift, as it reveals a gradual and broad transition. For this reason, it is evidenced in Figure 2b with a circle.

Moreover, the fusion enthalpy value of PA12 was 51.4 J/g, allowing estimation of the crystallization degree of the semicrystalline PA12 powder following Equation (1) [17,18], considering 100% crystalline PA12 polymer as 209 J/g based on the available literature [36,37]. The value of the degree of crystallinity percentage calculated for PA12 powder was 24.6%, as reported in Table 2.

The evaluation of the polymer phase transitions plays an important role in determining the sintering window of PA12 powder for SLS applications, which is defined as the temperature range between crystallization and melting point onset temperatures, as already reported in literature [17,18]. By analyzing the DSC curves, the sintering window for PA12 was found in the range between 150 °C and 170 °C. This result suggests that the PA12 powder can be successfully processed by SLS technique, as expected.

Moreover, the average true density of the PA12 powder was evaluated using gas pycnometer, and it was found equal to 1.05 ± 0.002 g/cm^3^, in line with the declared value in the supplier datasheet of the PA12 commercial powder, 1.00 g/cm^3^.

### 3.2. 3D Printing of Chess-like Pieces by SLS

Different printing tests were carried out to assess the effective printing parameters and printability of PA12 polymer powder by SLS. Then, several specimens were successfully 3D-printed via SLS, starting from simple geometries, like squares and rectangles, up to the realization of more complex geometries, such as typical chess-like structures, as illustrated in Figure 3.

The obtained parts reveal a well-detailed shape with a good alternation of full and empty spaces and thin curved walls, very difficult to print. The specimens are well-consolidated and show very complex structures, as can be observed in Figure 4.

The pictures reveal the realization of detailed chess-like pieces obtained by the deposition layer by layer of PA12 powder from 350, 400, and 480 layers for the pawn Figure 4a, the rook Figure 4b, and the knight Figure 4c, up to 500, 550, and 650 layers to produce the bishop Figure 4d, the queen Figure 4e, and the king Figure 4f, respectively.

The chess-like pieces are all characterized by a high level of detail and good dimensional stability, without macroscopic defects. This underlines that the selected printing parameters are optimized for obtaining very good printing definition and accuracy using SLS as a 3D printing technique.

### 3.3. Density Tests of PA12 Chess-like Pieces

The density values of the printed chess-like pieces were experimentally determined by using the buoyancy method based on the Archimedes principle. It is based on the difference in buoyancy of an object measured in air and submerged into a liquid. In the present work, isopropyl alcohol was employed as a reference liquid with known density. The advantages of the Archimedes method are that it is non-destructive, relatively inexpensive, and quick. However, it can be used to estimate a global density value relative to the reference liquid. Moreover, the density must be compared to the material’s nominal reference density [17,18,38,39,40,41]. Table 3 reports the values of density and porosity of all the 3D-printed PA12 chess-like pieces. The buoyancy method led to determining the average densities, considering the closed and total porosity estimated by using Equations (4) and (3), respectively, and the porosity by Equation (2).

The density values of the closed porosity are in the range between 0.994 ± 0.002 and 1.012 ± 0.003, and 0.989 ± 0.005 and 1.009 ± 0.006 for the total porosity, for the bishop and the rook, respectively. This leads to evaluating a percentage of total porosity of 4.4% for the pawn and the rook, and up to 6.3% for the bishop. These results give a clear indication about the good densification and definition of the PA12 samples obtained via SLS.

### 3.4. Morphology of PA12 Printed Parts

The microstructure of the 3D specimens realized by using SLS was investigated by means of SEM analysis, as reported in Figure 5. The polymeric sample appears well-sintered, confirming that the printing parameters set for the manufacturing process are optimized for SLS. Only small parts of the sample (evidenced by the white arrows) show the presence of particles not completely sintered, mainly for the side part, and this is probably due to the intrinsic characteristics of the polymer, which contains amorphous regions (i.e., crystallinity of about 25% as reported in Table 2) that can cause low coalescence between particles and low adhesion among layers during the printing process [19].

### 3.5. Thermal Properties of PA12 Printed Parts

Thermal properties of the 3D-printed chess-like pieces were evaluated by means of TG and DSC analyses. Figure 6a reports the thermograms and the DTG curves of the samples obtained by SLS and evidences that the sintering process does not affect the thermal stability of the polymer, which remains almost the same.

The thermal degradation for the printed PA12 samples involves a single step with a maximum peak located at around 436 °C, in line with the value of the neat PA12 powder, located at 438 °C (see Table 2).

By examining the DSC thermograms reported in Figure 6b, it is also possible to see that the 3D-printed specimens present single crystallization and melting phenomena, with peaks at 145 °C and 175 °C, respectively. Moreover, the degree of crystallinity was found at 25%. The PA12 printed specimens and pure powder have the same thermal behavior, which appears thermally stable after the sintering process. This result evidences that the printing process does not affect the thermal stability or the thermal properties of the PA12, which remain almost the same, as already seen for other thermoplastic powders processed via SLS [17,18,19]. No degradation of the powder occurs during the sintering process. The temperature range chosen in the range between 166 and 170 °C seems to be valuable for the bed powder in the SLS printer.

### 3.6. Microcomputed Tomography of PA12 Printed Parts

X-ray microcomputed tomography (µ-CT) has been considered a powerful, non-destructive, and useful tool to investigate the internal porosity of specimens with complex geometry resulting from SLS processes thanks to its capability of providing a complete analysis of size, shape, and distribution of pores/defects, focusing on how the printing process parameters, such as building direction, laser power and speed, and powder bed characteristics, can affect the pore formation [42,43,44].

During a tomographic scan, a series of 2D X-ray projections can be acquired at different angles as the sample, placed on a rotating stage and irradiated with an X-ray beam, rotates around the rotary axis. These projections are then used to create a 3D voxel (volumetric pixel) model of the sample by means of a back-projection algorithm [45].

In µ-CT, the pore distributions from several tomographic images, generally developed by X-ray, are combined to determine the total porosity of the samples.

The results of the porosity values calculated from µ-CT for the bishop and knight printed specimens are reported in Table 4.

The total porosity is in line with what is expected by the SLS fabrication technique [17,42], and it is very similar between the two pieces, the bishop and the knight. In both cases, the open porosity accounts for most of the total. This observation can be better understood by investigating the pore size distribution of the specimens.

As can be seen, the results of porosity measurements by µ-CT method are higher with respect to the ones previously determined by the Archimedes’ method, where the total porosity reached 4.6% and 6.3% for the knight and the bishop, respectively (see the Section 3.4).

These differences can be obviously attributed to the intrinsic characteristics of the two different methods. In fact, µ-CT can be considered a high-level technique for porosity evaluation. It is based on an expensive and time-consuming process of data acquisition and interpretation, mainly due to the need for high-resolution rendering and high-performance imaging machines [43,44,45,46]. In contrast, cost-effectiveness, ease of use, and feasibility are the main reasons that Archimedes’ method is favorable for the determination of the porosity of solid pieces with irregular shape.

Moreover, localized porosities due to process instabilities cannot be assessed individually, and the internal defects should be closed so as not to allow fluid to infiltrate the submerged part. For these reasons, its results are less accurate, and the test accuracy depends on how precise the weight or volume changes are measured by the operator [43,44,45,46].

Figure 7 shows that most of the pores have a small dimension for both the 3D-printed chess pieces, included in a range between 20 and 100 µm, probably due to the interspace among unconsolidated particles.

As is possible to see in Figure 8, larger pores, around 200 µm, were detected, but in an extremely low amount. The porosity is not homogeneously distributed either along the building direction or on the section of the objects, and it can be noticed that the surface and the core have distinct porosity features.

Moreover, on the surface, there is a poorly sintered skin, spanning from about 300 µm up to more than 1 mm thick, which is formed by partially unmelt particles in the heat-affected areas and constitutes most of the porosity, mainly as open one. The core of the samples is usually well-consolidated with only a few closed pores.

However, some defective layers can be observed in Figure 8a, sections 1–4, and Figure 8b, sections 1–3, particularly in proximity to section changes. This can explain the differences in the amount of open and closed porosity detected in both samples. The skin porosity is of open type, while the closed porosity is typical of the inner and denser part of the objects.

By analysing the porosity distribution, it is possible to observe three main things: larger sections correspond to lower porosity, as visible in Figure 8a—sections 1–6 and Figure 8b—sections 3–6; decreasing section’s areas results in increased porosity while increasing section’s areas improves consolidation (Figure 8a—section 4 and Figure 8b—section 1); building grooves or joining self-standing elements leads to highly defected areas (Figure 8a—section 5, Figure 8b—section 5).

Larger sections are less porous than smaller ones since the skin layer accounts for a lower amount of the total area and the inter-layer time is higher, meaning that the time between the scanning of a certain point and the recoating phase plays an important role in the final porosity, as already observed by Pavan et al. [43,44]. Larger areas are associated with higher inter-layer times, allowing for longer sintering and consolidation times before a new and cold layer of particles is deposited.

Complex details, such as the groove of the bishop’s head and the muzzle of the knight, are critical concerning the porosity. In fact, in the proximity of the lower portion of the groove (Figure 8a—section 5) or when the nose joins with the knight’s main body (see Figure 8b—section 5), highly defective areas were observed. It can be hypothesized that the temperature of the PA12 polymer powder under the recoated layer plays a role in this. In the case of the bishop, the lower part of the groove stays on top of the sintered and hot material, and therefore the heat flux can favour a partial consolidation of the powders even without being scanned by the laser. On the other hand, when the nose of the horse starts to attach to the main body, the first layers are built on top of the unsintered PA12 powder, at the bed temperature, which cannot provide additional heat, besides the heat of the lasers, to improve the consolidation of the polymer. The temperature of the underlying material can also play a role in the porosity of increasing or decreasing section areas. In fact, the main causes of these porosity changes, particularly at different sections of the 3D structures realized via SLS, may be related to variations in the processing conditions and material behavior during the manufacturing process. These factors may include temperature gradients, cooling rates, and the flow of the PA12 powder during processing. For example, in areas with slower cooling, the polymer may experience incomplete consolidation, leading to higher porosity. Additionally, the complex geometry of the part can contribute to localized variations in porosity. These factors are particularly relevant to our study and help explain the observed porosity differences across various sections of the chess-like pieces.

To our best knowledge, this is the first time that the effect of complex shapes on porosity has been reported for SLS manufacturing processes of PA12 powder.

Previous studies have evaluated the effects of processing parameters in terms of laser energy density, scanning pattern, and building orientation only on simple shapes, such as cubes [44], thin disks [45], or cylinders [46].

Our results highlight that to improve the overall quality of complex SLS printed objects, in terms of densification and printing precision, it is necessary to carefully evaluate how the last printed layers affect the consolidation of the in-building one through heat flow and the role of the laser in re-heating previously printed layers by superimposing scanning areas.

Furthermore, X-ray micro-computed tomography scan data were used to study the geometrical accuracy of some PA12 printed parts, which represents a primary issue in 3D printing applications.

The analysis uses the variance of the scan model from the STL file generated by the images’ reconstruction to measure the accuracy of the final 3D-printed chess-like pieces, by analysing the discrepancies arising from the 3D printing and the scanning procedures and addressing any possible structural defects. Since the reference system between CAD and scan is not usually the same, the CAD and scan data should be aligned to perform a comparison [47,48,49]. The pre-alignment and best-fit functions were used to perform the alignment. The alignment procedure consists of minimizing the distance between scan and CAD data, applying a roto-translation matrix to the scan data.

Figure 9 reports the CAD models and the alignments for the 3D-printed chess-like pieces: bishop Figure 9a,b and the knight Figure 9c,d, respectively.

To compare them, the software calculates the distance between the scan data and the CAD data and gives back a colored map which allows visualization of the displacement of the scan data with respect to the original CAD file used to print the piece.

Color maps, which give tolerance-based deviations, are used to show model differences from the reference model. The creation of a colored map for the analysis of the deviations is a good graphical way to have visual feedback on the quality of the produced part.

In the present study, tolerance for the scanned models was set from −0.3 mm to +0.3 mm. Figure 10 reports the colored maps of two different chess-like pieces analyzed: bishop in Figure 10a and knight in Figure 10b. These maps illustrate the differences of the 3D scan data from the CAD model originally used to realize via SLS the chess-like pieces.

For the bishop, the average displacement is −0.03 mm with a standard deviation from the average of 0.53 mm. In contrast, the knight has an average displacement of 0.20 mm and a standard deviation from the average of 0.40 mm.

Table 5 reports the comparison between the average displacements and the standard deviations for the bishop and the knight.

Looking at the coloured map of the bishop illustrated in Figure 10a, it is possible to see that most of its surface is light and dark green, which means that the displacement is quite null as numerically confirmed by the average distance, but the high standard deviation (±0.53 mm) means that there are a lot of surfaces in positive or negative displacement with respect to the CAD file.

This is probably due to the presence of locations in the analysed area that are difficult to scan because of the shadows. The changes in the cross-section of the bishop, due to its complex shape, results in a critical issue and can create zones, red coloured, in which the distance is positive, indicating a high deviation with respect to the initial CAD model. Additionally, some small zones are blue due to the overhanging surfaces, meaning that the displacement is negative and the discrepancy with respect to the CAD model is high.

Figure 10b also shows the coloured map of the knight, revealing that almost its entire surface is green. As well as for the bishop, the changes in the cross-section for the knight represent a critical issue. In this case, it leads to the creation of a zone in which the distance is positive, evidencing a high discrepancy with respect to the CAD model. The high value of displacement can be attributed to a higher complexity in the shape and geometry of the knight with respect to the bishop. These findings indicate that the 3D-printed models derived from the reconstruction of the 3D images of the bishop and knight fall within acceptable limits of tolerance, underscoring the good accuracy and resolution of the PA12 chess-like pieces realized by SLS 3D printing.

## 4. Conclusions

3D-printed parts with complex geometries, such as chess-like pieces, were successfully manufactured by means of selective laser sintering using PA12 as polymeric powder. The powder printability was studied by identifying the best parameters to optimize the printing process considering PA12 flowability, sintering window, thermal stability, and morphology. Thermal analyses, performed by TG and DSC, revealed that the sintering process occurring during the printing process does not have effects on the thermal degradation of the PA12 powder, which led to evaluating the crystallinity degree of the polymer powder and to defining the ideal temperature range for obtaining highly densified and complex specimens via SLS.

These findings were confirmed by SEM and porosity tests, which led to investigating the microstructure of the 3D-printed chess-like pieces. The PA12 powder particles seem well-sintered, underscoring once again a good choice of printing parameters.

Furthermore, an in-depth analysis of the porosity was carried out by using X-ray micro-computed tomography, which allows to have indications on the size and shape of voids and defects of the two selected 3D-printed chess-like objects, like bishop and knight, correlating them with the printing parameters.

The results evidenced the presence of few pores with tiny dimensions, probably due to the interspace among unconsolidated PA12 particles, and different porosity features between the surface and the inner part of the complex 3D-printed pieces.

The total porosity is higher than that found by using the buoyancy method and reaches values of about 9.2%, mainly due to the higher resolution of the μ-CT technique, which can analyse even localized porosities in specific parts of the sample. However, the results are in line with the requirements for the SLS 3D printing technique.

Finally, the dimensional accuracy of the bishop and knight chess pieces was investigated by μ-CT scan, evaluating the differences between the digital recreation of the selected printed parts made by 3D scan and the original CAD model used for the SLS printing process.

The overall results clearly indicate that the chess-like pieces realized via SLS using PA12 powder show a high level of structural complexity accomplished by a very good definition and dimensional accuracy.

## Figures and Tables

**Figure 1 polymers-16-03526-f001:**
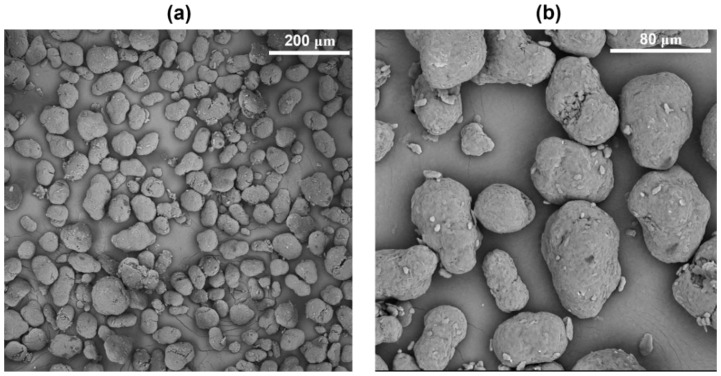
SEM micrographs of PA12 powder particles at different magnifications, 300× (**a**) and 1000× (**b**), respectively.

**Figure 2 polymers-16-03526-f002:**
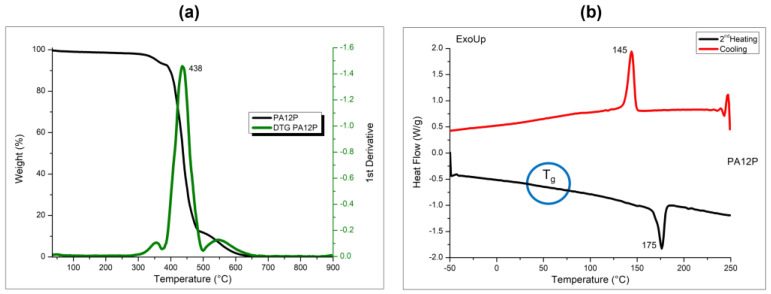
TG and derivative TG (DTG) curves (**a**) and DSC curves (**b**) of PA12 powder.

**Figure 3 polymers-16-03526-f003:**
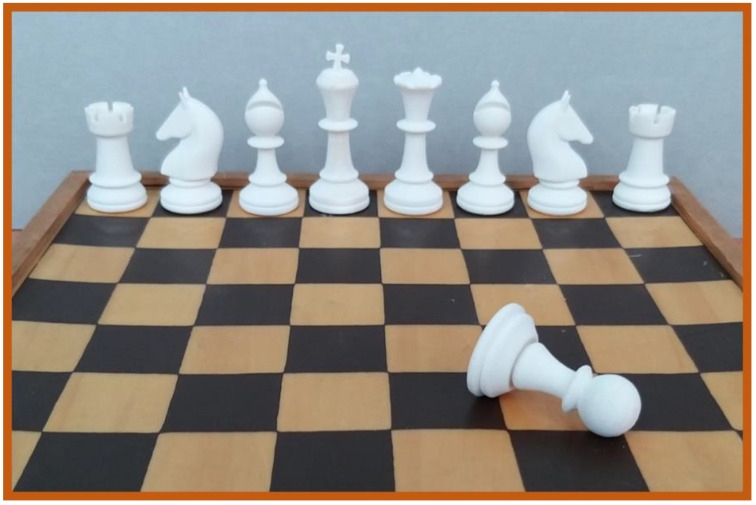
3D printed chess-like pieces realized via SLS using PA12.

**Figure 4 polymers-16-03526-f004:**
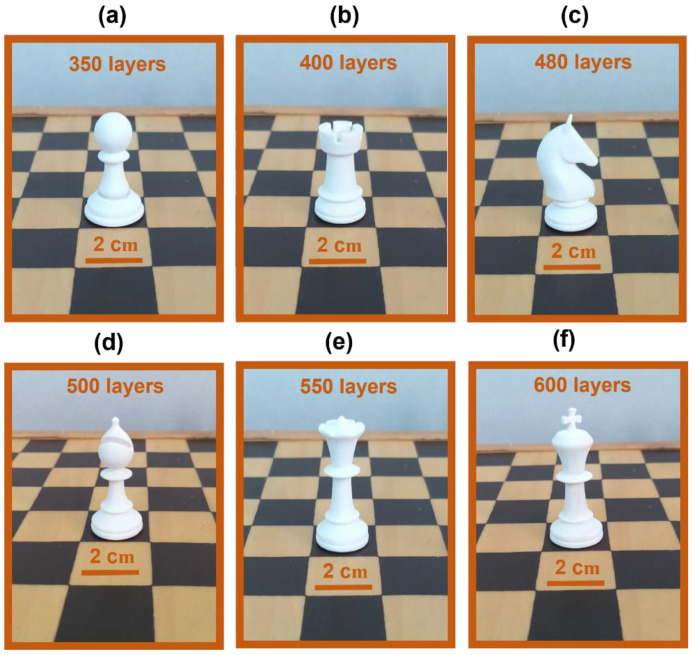
Details of chess-like pieces realized via SLS: pawn (**a**), rook (**b**), knight (**c**), bishop (**d**), queen (**e**), and king (**f**).

**Figure 5 polymers-16-03526-f005:**
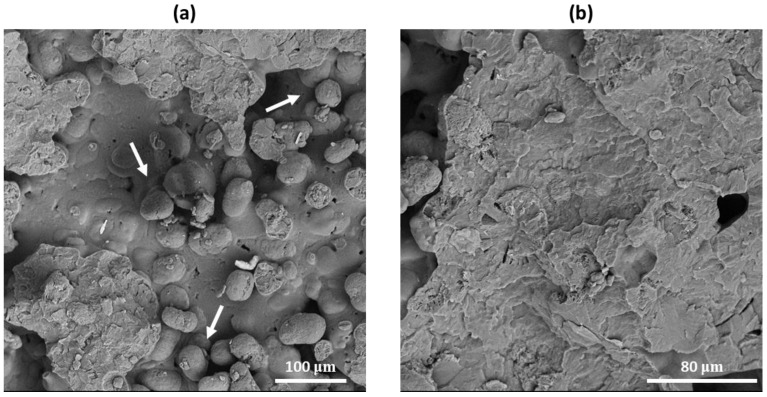
SEM micrographs of the fracture surface of 3D-printed PA12 chess pieces at 500 (**a**) and 1000× (**b**), respectively.

**Figure 6 polymers-16-03526-f006:**
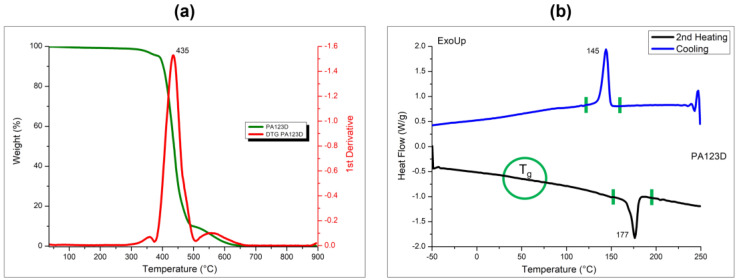
TG and DTG curves (**a**) and DSC (**b**) curves of PA12 3D-printed samples.

**Figure 7 polymers-16-03526-f007:**
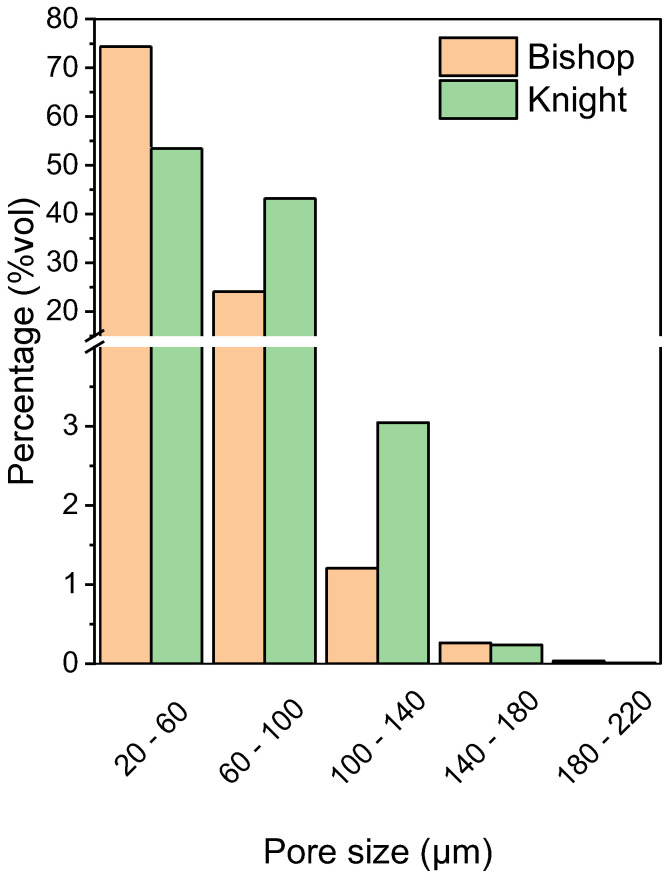
Pore size distribution as volume percentage for some 3D-printed specimens: bishop and knight.

**Figure 8 polymers-16-03526-f008:**
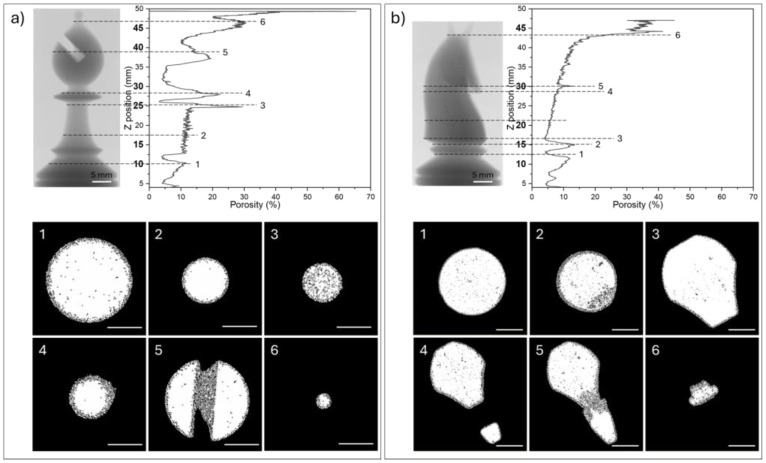
X-ray projection and distribution of the porosity along the building axis of the bishop (**a**) and the knight (**b**) with representative binarized cross-section images (scale bar: 5 mm). The porosity is the relative porosity for each cross-section. Images are numbered according to their position along the scanned piece.

**Figure 9 polymers-16-03526-f009:**
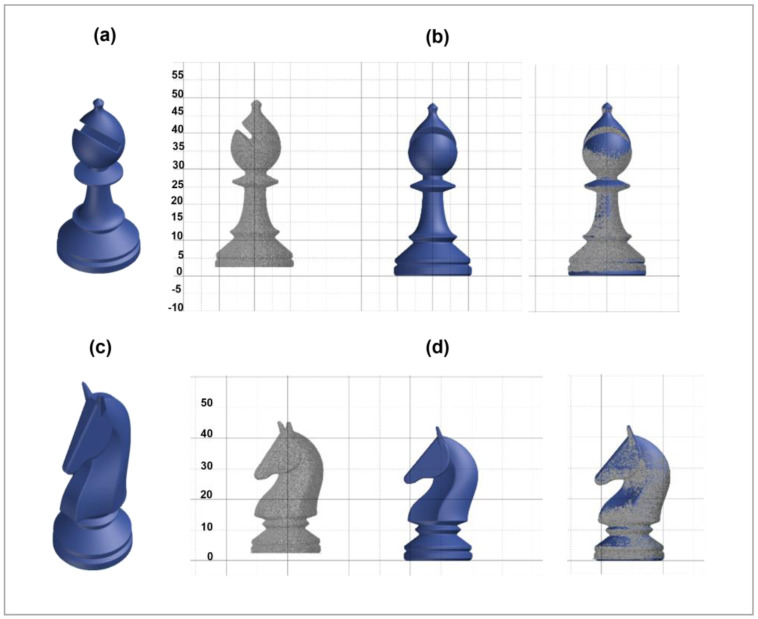
CAD models and 3D scan alignments of the bishop (**a**,**b**), and the knight (**c**,**d**).

**Figure 10 polymers-16-03526-f010:**
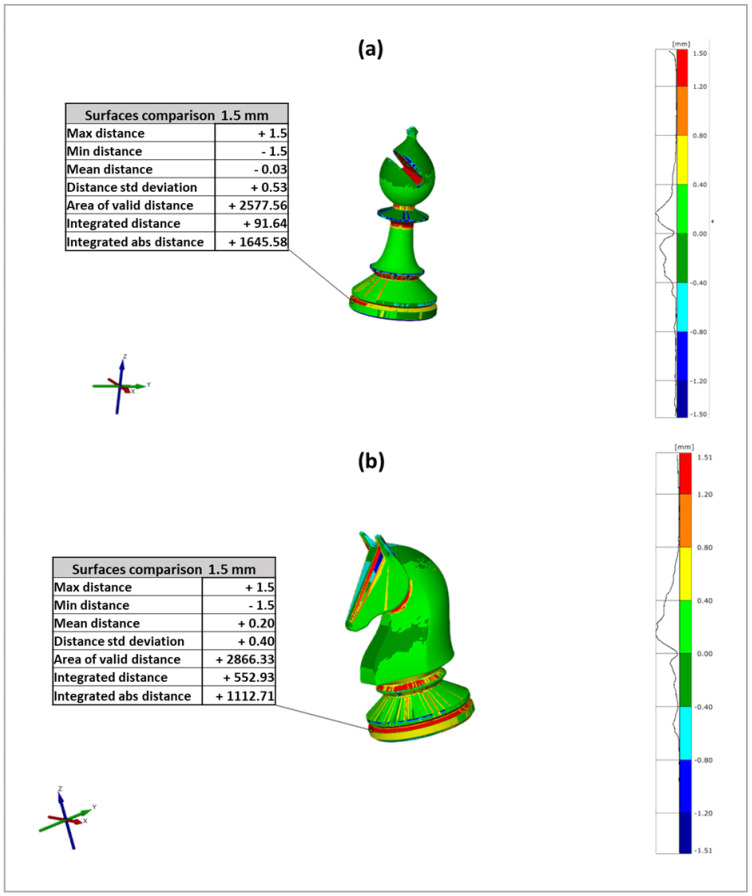
Colored maps of the bishop (**a**) and the knight (**b**) printed via SLS.

**Table 1 polymers-16-03526-t001:** PA12 particle size distribution, diameter, and circularity evaluated by morphological imaging.

Sample	Diameter (µm)	Aspect Ratio	Circularity
PA12P	D [n, 0.1]: 9.5	D [n, 0.1]: 0.5	D [n, 0.1]: 0.7
	D [n, 0.5]: 39.2	D [n, 0.5]: 0.7	D [n, 0.5]: 0.9
	D [n, 0.9]: 69.5	D [n, 0.9]: 0.9	D [n, 0.9]: 0.9
	D [n, 0.1]: 9.5	D [n, 0.1]: 0.5	D [n, 0.1]: 0.7

**Table 2 polymers-16-03526-t002:** Thermal properties of PA12 powder and 3D-printed samples.

Sample	T_ONSET_ ^A^(°C)	T_MAX DEG_ ^B^ (°C)	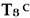 (°C)	T_□_(°C)	ΔH_□_ ^C^(°C)	T_C_ ^C^(°C)	X_C_ ^D^(%)
PA12P	356	438	60	175	51.4	145	24.6
PA123D	390	436	55	177	52.8	145	25.0

^A^ Onset temperature determined by TG in air; ^B^ Maximum degradation peak temperature determined by DTG curves; ^C^ Transition temperatures determined by DSC in nitrogen; ^D^ Calculated using the Equation (1).

**Table 3 polymers-16-03526-t003:** Density and porosity values of the printed PA12 chess-like pieces.

Sample	Density (Closed Porosity) (g/mL)	Density (Total Porosity) (g/mL)	Porosity (%)
Pawn	1.011 ± 0.003	1.009 ± 0.003	4.4
Rook	1.012 ± 0.003	1.009 ± 0.006	4.4
Knight	1.010 ± 0.002	1.007 ± 0.003	4.6
Bishop	0.994 ± 0.002	0.989 ± 0.005	6.3
Queen	1.009 ± 0.002	1.006 ± 0.003	4.7
King	0.998 ± 0.001	0.994 ± 0.004	5.8

**Table 4 polymers-16-03526-t004:** Total porosity, open and closed porosity of some 3D-printed chess pieces evaluated by micro-CT.

Sample	Total Porosity (%)	Open Porosity (%)	Closed Porosity (%)
Bishop	9.2	8.2	1.0
Knight	8.8	7.7	1.3

**Table 5 polymers-16-03526-t005:** Results of the comparisons between CAD models and scan data.

Sample	Average Distance 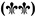	Max Distance 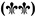	Min Distance 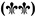	Std Deviation 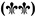
Bishop	−0.03	1.5	−1.5	0.53
Knight	0.20	1.5	−1.5	0.40

## Data Availability

Research data are available if request.

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
