# Peer review of "Chess-like Pieces Realized by Selective Laser Sintering of PA12 Powder: 3D Printing and Micro-Tomographic Assessment"

_polymers, 2024, doi:10.3390/polym16243526_

Round 1

Reviewer 1 Report

Comments and Suggestions for Authors

Review report

I have gone through the manuscript titled “Chess-like pieces realized by selective laser sintering of PA12 powder: 3D printing and micro-tomographic assessment” by Colucci et al.

The paper presents an analysis of complex shaped 2D printed objects made of PA12. The micro-tomography analysis of the printed objects was reported. Also, the powder characterization data were presented. The manuscript is well written and adds value to the field of additive manufacturing. The figures and analyses are also properly described. However, a few issues, as indicated below are to be addressed before considering the manuscript for publication.

1) Overall, the introduction is fine. However, there are too many short paragraphs which may be consolidated.

2) Line no. 82, on page-2: too many references are grouped together. It will be better if a a little more explanation with specific reference is provided.

3) The last paragraph of section 2.1 lists all the experimental parameters. How were these parameters optimized. A little more detail with suitable reference (if any) will be better. This will also help in brining out the good amount of work behind the results presented in this paper.

4) Please check the subscripts in Eq. 3 denominator.

5) Also, the symbol for density is not matching in the equations and in the text. Please check.

6) Line no. 188, page-4: “potato-shaped” may not be a proper scientific expression. Please reconsider.

7) Fig. 2b: why 2nd cooling data was used?

8) The glass transition at 60 deg. C is not clear from the DSC data. Additional graphs (may be in the set of Fig. 2b) will be better.

9) In the title of section 3.2 D printing shall be 3D printing.

10) Line 245 on page 7: the word layers shall come after 650.

11) Line 271 on page 8: Is there a standard on acceptable level of porosities? Also, how the estimated porosity percentage tally with literature?

12) In Fig. 5, it will be better if an arrow is used to indicate portions which have been not sintered.

13) Fig. 6b, the endset and onset temperature are to be indicated.

14) The discussion on the porosities from micro-CT is unnecessarily long. Please summarize in 2-3 paragraphs.

15) The data on the variation of 3D scan and CAD data is nicely represented.

16) Please correct the distance units in Table 5.

17) The conclusion is too long. Please rewrite briefly with the key insights obtained. Please do not include long sentences. Also, the last line of the conclusion section is a future goal, as this was not studied in the preset paper.

Author Response

Reviewer Comments

Reviewer 1

I have gone through the manuscript titled “Chess-like pieces realized by selective laser sintering of PA12 powder: 3D printing and micro-tomographic assessment” by Colucci et al.

The paper presents an analysis of complex shaped 2D printed objects made of PA12. The micro-tomography analysis of the printed objects was reported. Also, the powder characterization data were presented. The manuscript is well written and adds value to the field of additive manufacturing. The figures and analyses are also properly described. However, a few issues, as indicated below are to be addressed before considering the manuscript for publication.

  • Overall, the introduction is fine. However, there are too many short paragraphs which may be consolidated.

We thank the reviewer for her/his suggestion. We modified the introduction in the revised version of the manuscript to be clearer and avoid too many short sentences.

  • Line no. 82, on page-2: too many references are grouped together. It will be better if a little more explanation with specific reference is provided.

We acknowledge the reviewer’s comment and, accordingly to her/his suggestion, we modified the text by adding more explanation on specific references in the revised version of the manuscript.

“Many studies have examined the relation between the process parameters, such as laser power, scan speed, layer thickness, and preheating temperature, and the part properties in terms of tensile strength, modulus, elongation at break, and density. The aim was to extend process knowledge on SLS of polymers, increasing the process control [28-31]. Other authors have explored how the process parameters strongly influence the structures containing voids correlating this aspect with their final properties. As pores are generally considered as critical defects, the research seeks to understand how the porosity and the pores formation can influence the morphology and the mechanical properties of the printed parts obtained via SLS [32-35].”

  • The last paragraph of section 2.1 lists all the experimental parameters. How were these parameters optimized? A little more detail with suitable reference (if any) will be better. This will also help in bringing out the good amount of work behind the results presented in this paper.

According to the reviewer’s suggestion on this aspect, we modified the text in the revised version of the manuscript by inserting a better explanation on how we set the printing parameter for the SLS processing of the PA12 powder.

The optimization of the printing parameters was done by means of a comprehensive experimental testing procedure. Initially, the printability of the PA12 powder was assessed starting from printing parameters reported from previous literature on similar works. Then, a systematic optimization of the relevant settings for the realization of the chess-like pieces was carried out. The printing parameters were changed for every single print until the best resolution details were obtained.

  • Please check the subscripts in Eq. 3 denominator.

We thank the reviewer for her/his suggestion. We carefully checked the equation 3 and 4 in the manuscript by doing the requested corrections.

  • Also, the symbol for density is not matching in the equations and in the text. Please check.

According to the reviewer’s suggestion, we made all the changes necessary to align the text in order that the symbols matched with the text and the equations reported in the whole manuscript.

  • Line no. 188, page-4: “potato-shaped” may not be a proper scientific expression. Please reconsider.

      We thank the reviewer for her/his suggestion. We took the opportunity to rewrite the  sentence with a proper language to improve the quality of the manuscript. The expression “potato-shaped particles” was changed with a more correct “irregular elongated-shaped particles”.

  • 2b: why 2ndcooling data was used?

We thank the reviewer for her/his suggestion. We took the opportunity to better explain this point. Figure 2 (b) reports the DSC curves, performed in nitrogen, relative to the single cooling run and the second heating run. The first heating run was done to eliminate the thermal history of the polymer sample. The thermal transitions reported in the paper were evaluated based on the results of the cooling and second heating cycles.

  • The glass transition at 60 deg. C is not clear from the DSC data. Additional graphs (may be in the set of Fig. 2b) will be better.

We agree with the reviewer regarding the fact that the Tg at 60 °C is not clear from DSC curve. However, this is a typical behavior of semi-crystalline polymers, like PA12. The glass transition may not result in a sharp, well-defined baseline shift, as it can be a gradual and broad transition. This makes it difficult to pinpoint a clear Tg in the DSC graph. We modified the DSC plot to put in evidence the range where the Tg value is located by inserting a circle in the thermogram.

      9) In the title of section 3.2 D printing shall be 3D printing.

We thank the reviewer for her/his suggestion. We modified the title of section 3.2 by correcting the typo.

  • Line 245 on page 7: the word layers shall come after 650.

We thank the reviewer for her/his suggestion. We took the opportunity to correct the sentence, as suggested.

  • Line 271 on page 8: Is there a standard on acceptable level of porosities? Also, how the estimated porosity percentage tally with literature?

We thank the reviewer for her/his suggestion. We took the opportunity to clarify this point. In SLS, there is not a well-defined level of porosity due to the intrinsic characteristics of the process. However, several guidelines, research studies, and industry practices help in understanding the acceptable levels of porosity for SLS components that strictly depend on the final application, material, and required final mechanical properties. Porosity typically ranges from 1% to 5% for high-quality functional parts. Higher porosity up to 10% can also be acceptable in those applications where the mechanical strength is less critical, such as prototypes or non-structural components, for example for biomedical applications. The porosity percentage values reported for the chess-like pieces realized via SLS in our paper align with literature values, but variations can occur depending on the material, process parameters, and post-processing treatments eventually used [17,42].

12) In Fig. 5, it will be better if an arrow is used to indicate portions which have been not sintered.

We thank the reviewer for her/his suggestion. We modified the Figure 5 by adding some white arrows to evidence the portions of PA12 powder not well-sintered within the specimen.

13) Fig. 6b, the endset and onset temperature are to be indicated.

We thank the reviewer for her/his suggestion. We modified the Figure 6 (b) by adding the onset and endset temperatures on the DSC curves.

14) The discussion on the porosities from micro-CT is unnecessarily long. Please summarize in 2-3 paragraphs.

We thank the reviewer for her/his comments on the micro-CT description. According to her/his suggestion we modified the paragraph 3.5 in the revised version of the manuscript in order to be more concise and clearer.

15) The data on the variation of 3D scan and CAD data is nicely represented.

We really appreciate the reviewer for her/his comment. Thanks.

16) Please correct the distance units in Table 5.

We thank the reviewer for her/his suggestion. We modified the distance units of Table 5 in the revised version of the manuscript, as requested.

17) The conclusion is too long. Please rewrite briefly with the key insights obtained. Please do not include long sentences. Also, the last line of the conclusion section is a future goal, as this was not studied in the preset paper.

We appreciate the reviewer for her/his comment. We rearrange the conclusions to be more concise and improve the flow of the text in the revised version of the manuscript

Reviewer 2 Report

Comments and Suggestions for Authors

Colucci et al present SLS based approach to print complex structure such as chess pieces and analyze porosity of the structures via different approach to understand the structure.

The current manuscript requires minor editing/clarifications before acceptance and the comments are provided below.

1.      Authors mention the glass transition temperature for the polyamide at Page 6, line 212. What is the significance of Tg in the current context. The addition of 1-2 lines addressing this point will be helpful.

2.      Can authors clarify Table 1.

3.      What is the leading cause of porosity changes, especially at the different sections of the structure. Can authors clarify this.

4.      How does these changes/variations in the porosity affect the mechanical strength of the structure?

5.      Units are missing from Table 5.

6.      Sentence correction is needed in line 233.

Author Response

Reviewer 2

Colucci et al present SLS based approach to print complex structure such as chess pieces and analyze porosity of the structures via different approach to understand the structure.

The current manuscript requires minor editing/clarifications before acceptance and the comments are provided below.

  1. Authors mention the glass transition temperature for the polyamide at Page 6, line 212. What is the significance of Tg in the current context? The addition of 1-2 lines addressing this point will be helpful.

We thank the reviewer for her/his comment. The glass transition temperature (Tg) of PA12 is significant in our study because it provides insight into the thermal behavior and mechanical properties of the material. The Tg represents the temperature at which the polymer transitions from a rigid, glassy state to a more flexible, rubbery state. In the context of our research, we mentioned Tg to highlight the thermal properties of PA12, which is relevant for understanding its final performance in different temperature environments, like within the 3D printer. Moreover, the value of Tg can influence factors like processability and polymer behavior under stress, which are critical for SLS applications described in our study. These comments were introduced in the revised version of the manuscript according to her/his suggestion.

  1. Can authors clarify Table 1.

       We thank the reviewer for the request for this clarification. We took the opportunity to better explain which results are reported in Table 1 in the revised version of the manuscript.

      Table 1 summarizes the data evaluated by granulometric analysis of PA12 particles in terms of the diameter, aspect ratio, and circularity to determine the distribution of particle sizes within the powder, considering that particles with dimensions lower than 100 microns can be useful for SLS applications.

  1. What is the leading cause of porosity changes, especially at the different sections of the structure? Can authors clarify this.

We thank the reviewer for the request for these clarifications. We took the opportunity to better explain this aspect in the revised version of the manuscript.

The main causes of porosity changes, particularly at different sections of a 3D structure realized via SLS, is often related to variations in the processing conditions and material behavior during the manufacturing process. These factors can include temperature gradients, cooling rates, and the flow of powder during the processing. For example, in areas with slower cooling, the polymer powder may experience incomplete consolidation, leading to higher porosity in those sections. Additionally, the complex geometry of the parts can contribute to localized variations in porosity. These factors are particularly relevant to our study and help explain the observed porosity differences across various sections of the chess-like pieces.

  1. How does these changes/variations in the porosity affect the mechanical strength of the structure?

Thank you for your follow-up question. The variations in porosity within a 3D part produced by SLS can significantly affect its mechanical strength, because porosity acts as a type of defect within the material, and directly influence the final material's properties. For example, areas with higher porosity led to the formation of weak points that can initiate cracks or fractures under stress, causing a reduction in tensile and compressive strength or again voids can act as stress concentrators, which can accelerate crack propagation and lead to premature failure of the material.

  1. Units are missing from Table 5.

      We thank the reviewer for her/his suggestion. We took the opportunity to check all the units in the whole manuscript.

  1. Sentence correction is needed in line 233.

       We thank the reviewer for her/his suggestion. We took the opportunity to make all the necessary correction into the revised version of the manuscript.

Round 2

Reviewer 1 Report

Comments and Suggestions for Authors

The authors have provided suitable response to the review comments and made the necessary corrections to the manuscript. Because of this, the quality of the manuscript has been improved and the manuscript is now suitable for publication.